EMBO
Molecular Medicine

# Development of broad-spectrum human monoclonal antibodies for rabies post-exposure prophylaxis

Paola De Benedictis[1,†], Andrea Minola[2,†], Elena Rota Nodari[1], Roberta Aiello[1], Barbara Zecchin[1], Angela Salomoni[1], Mathilde Foglierini[3], Gloria Agatic[2], Fabrizia Vanzetta[2], Rachel Lavenir[4], Anthony Lepelletier[4], Emma Bentley[5], Robin Weiss[6], Giovanni Cattoli[1], Ilaria Capua[1], Federica Sallusto[3], Edward Wright[5], Antonio Lanzavecchia[3,7], Hervé Bourhy[4] & Davide Corti[2,3,*]

## Abstract

Currently available rabies post-exposure prophylaxis (PEP) for use in humans includes equine or human rabies immunoglobulins (RIG). The replacement of RIG with an equally or more potent and safer product is strongly encouraged due to the high costs and limited availability of existing RIG. In this study, we identified two broadly neutralizing human monoclonal antibodies that represent a valid and affordable alternative to RIG in rabies PEP. Memory B cells from four selected vaccinated donors were immortalized and monoclonal antibodies were tested for neutralizing activity and epitope specificity. Two antibodies, identified as RVC20 and RVC58 (binding to antigenic site I and III, respectively), were selected for their potency and broad-spectrum reactivity. *In vitro*, RVC20 and RVC58 were able to neutralize all 35 rabies virus (RABV) and 25 non-RABV lyssaviruses. They showed higher potency and breath compared to antibodies under clinical development (namely CR57, CR4098, and RAB1) and commercially available human RIG. *In vivo*, the RVC20–RVC58 cocktail protected Syrian hamsters from a lethal RABV challenge and did not affect the endogenous hamster post-vaccination antibody response.

**Keywords** human monoclonal antibody; lyssaviruses; post-exposure prophylaxis; rabies

**Subject Categories** Immunology; Microbiology, Virology & Host Pathogen Interaction

## Introduction

Rabies virus (RABV) belongs to the family *Rhabdoviridae*, genus *Lyssavirus*, and causes acute encephalitis in mammals. Lyssaviruses are enveloped single-stranded (-) RNA viruses which have helical symmetry and display on the outer surface of the virion envelope the G protein, which is the target antigen of virus-neutralizing antibodies.

RABV is the first of fourteen lyssavirus species identified to date (Dietzgen *et al*, 2011), with an additional yet unclassified putative species named Lleida bat lyssavirus (LLEBV) (Arechiga Ceballos *et al*, 2013). According to their viral genetic distances, two major phylogroups have been defined: Phylogroup I includes the species RABV, European bat lyssavirus type 1 (EBLV-1) and type 2 (EBLV-2), Duvenhage virus (DUVV), Australian bat lyssavirus (ABLV), Aravan virus (ARAV), Khujand virus (KHUV), Bokeloh bat lyssavirus (BBLV), and Irkut virus (IRKV); Phylogroup II includes Lagos bat virus (LBV), Mokola virus (MOKV), and Shimoni bat virus (SHIBV). The remaining viruses, West Caucasian bat virus (WCBV) and Ikoma lyssavirus, (IKOV) cannot be included in either of these phylogroups and have been temporarily assigned to putative phylogroups III and IV, respectively (Bourhy *et al*, 1992, 1993; Amengual *et al*, 1997; Hooper *et al*, 1997; Badrane *et al*, 2001; Kuzmin *et al*, 2010; Marston *et al*, 2012). It is currently thought that infection with all lyssavirus species culminates in viral encephalitis clinically indistinguishable from that caused by RABV and ultimately results in human and animal deaths.

RABV is found almost ubiquitously worldwide in different animal reservoirs, with occasional spillover events to non-reservoir

1 FAO and National Reference Centre for Rabies, National Reference Centre and OIE Collaborating Centre for Diseases at the Animal-Human Interface, Istituto Zooprofilattico Sperimentale delle Venezie, Legnaro, Padua, Italy
2 Humabs BioMed SA, Bellinzona, Switzerland
3 Institute for Research in Biomedicine, Università della Svizzera Italiana, Bellinzona, Switzerland
4 Institut Pasteur, Unit of Lyssavirus Dynamics and Host Adaptation, National Reference Centre for Rabies, World Health Organization Collaborating Centre for Reference and Research on Rabies, Paris Cedex 15, France
5 Viral Pseudotype Unit, Faculty of Science and Technology, University of Westminster, London, UK
6 Division of Infection and Immunity, University College London, London, UK
7 Institute of Microbiology, ETH Zurich, Zurich, Switzerland
 *Corresponding author. Tel: +41 91 825 63 80; E-mail: davide.corti@humabs.ch
 †These authors contributed equally to this work

hosts, including humans. Although almost 100% fatal following the onset of symptoms, rabies can be controlled in the animal reservoirs through mass vaccination and prevented through the appropriate prophylactic treatment in humans exposed to the virus. Approximately 17 million people per year are treated after exposure to rabies, in most cases following a bite from an infected animal. Some 59,000 people are estimated to die each year, mainly in Africa, China, and India, and 50% of rabies cases worldwide occur in children (Fooks *et al*, 2014; Hampson *et al*, 2015). However, the true burden of rabies-related lyssaviruses in developing countries is unknown and largely under-diagnosed (Mallewa *et al*, 2007).

In humans, rabies prevention is achieved by either pre- or post-exposure prophylaxis. If exposed to RABV, post-exposure prophylaxis (PEP) is recommended to prevent the advancement of infection and thus the clinical disease; however, it must be administered as early as possible. According to the World Health Organization (WHO) (World Health Organization. 2013), PEP includes the first-aid treatment of the wound and the administration of the rabies vaccine alone or in combination with rabies immunoglobulin (RIG) for category II or III exposures, respectively. In particular, patients with category III exposures should receive RIG administered into or around the wound site and four to five doses of vaccine. Two types of RIG are currently available for PEP: human or equine rabies immunoglobulin (HRIG and ERIG, respectively). The dose of HRIG recommended by the WHO is 20 IU/kg body weight (corresponding to 20 mg/kg); for ERIG and F(ab′)2 products, the recommended dose is 40 IU/kg body weight. Higher doses of RIGs have been shown to reduce vaccine efficacy (Atanasiu *et al*, 1956, 1961, 1967; Archer & Dierks, 1968; Sikes, 1969; Cabasso *et al*, 1971, 1974; Wiktor *et al*, 1971; Cabasso, 1974). HRIG is widely used in developed countries and considered safer than ERIG. The high cost of HRIG and its limited availability hamper its wide use in resource-limited countries, particularly in Africa (Dodet and Africa Rabies Bureau (AfroREB) 2009). Moreover, vaccine and HRIG or ERIG do not confer protection against infection with all non-RABV lyssavirus species, and protection is thought to be inversely related to the genetic distance with the RABV vaccine strain (Brookes *et al*, 2005; Hanlon *et al*, 2005; Both *et al*, 2012). Thus, a search for a replacement to HRIG has been strongly encouraged by the WHO (World Health Organization, 2013). To this end, mouse and human monoclonal antibodies have been developed in the last decade, with two products in advanced clinical trials, namely CL184 (produced by Crucell, based on the combination of two antibodies called CR57 and CR4098, Bakker *et al*, 2005; Goudsmit *et al*, 2006) and RAB1 (produced by Mass Biologics and Serum Institute of India, based on a single monoclonal antibody, Sloan *et al*, 2007; Nagarajan *et al*, 2014). However, RABV isolates that are not neutralized by each of these monoclonal antibodies have been identified (Marissen *et al*, 2005; Kuzimina *et al*, 2013). These findings highlight the challenge to perform Phase 2 or 3 trials where the risk of monoclonal antibody-based PEP failures poses a serious ethical concerns. Indeed, for the lack of broad RABV coverage, the development of CL184 was recently halted, while RAB1 in still under Phase 2 or 3 development in India. Thus, in the selection and development of a safe and effective monoclonal antibody-based PEP of RABV infections, it is of paramount importance to identify neutralizing monoclonal antibodies that are able to recognize G protein sequences of RABV from all lineages. As previously described for other viral targets (Corti &

Lanzavecchia, 2013), the combination of two antibodies that bind to different antigenic sites on the RABV G protein and are able to broadly neutralize both RABV and non-RABV lyssavirus isolates will significantly reduce the risk of PEP failure.

# Results

## Selection of rabies vaccinees and isolation of potent RABV-neutralizing antibodies

In order to isolate broadly neutralizing antibodies against not only RABV isolates but also non-RABV lyssaviruses, sera from 90 RABV vaccinees were screened for the presence of high titers of antibodies that bind to the RABV (CVS-11 isolate) G protein by ELISA (Fig 1A) Of these, the 29 with the highest binding titers ($ED_{50} > 50$) were tested for their ability to neutralize a panel of 12 pseudotyped lyssaviruses representing RABV and non-RABV lyssaviruses isolates of phylogroup I, II, and III viruses (Fig 1B and Appendix Table S1). HRIG Berirab® was included as a reference. As expected, all samples neutralized, albeit with variable titers, the CVS-11 isolate (RABV). The neutralization profile of the other lyssavirus species varied considerably in all donors tested, but in a few cases, all species were neutralized. It was interesting to note that HRIG showed only modest activity against non-RABV phylogroup I species and no cross-reactivity with phylogroup II and III viruses.

Memory B cells from four vaccinees selected for the presence of serum antibodies capable of broadly neutralizing multiple lyssavirus species were immortalized with Epstein–Barr virus (EBV) and CpG, as previously described (Traggiai *et al*, 2004). Culture supernatants were then tested using a 384-well-based RABV (CVS-11) pseudotyped neutralization assay on BHK-21 cells. Five hundred

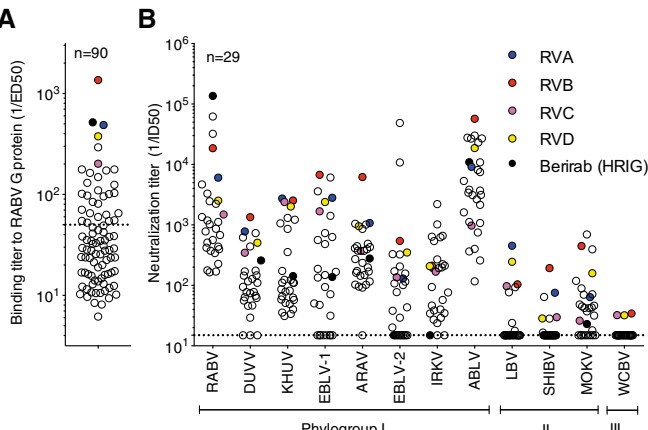

**Figure 1. Selection of RABV vaccinees with broadly reactive neutralizing antibodies.**

A  A panel of 90 sera were tested by ELISA for binding to RABV G protein. Shown are the $1/ED_{50}$ values.

B  Sera samples selected for high binding titers ($1/ED_{50} > 50$) values were tested for the presence of neutralizing antibodies against a panel of 12 pseudotyped viruses. $1/ID_{50}$ values are shown. Black circles indicate HRIG (Berirab®), and colored circles indicate the four donors selected for the memory B-cell interrogation.

human monoclonal antibodies were isolated for their ability to neutralize pseudotyped CVS-11 RABV. Twenty-one human monoclonal antibodies were selected for their high neutralizing potency against CVS-11 RABV pseudotyped virus, with $IC_{90}$ (concentration of antibody neutralizing 90% of viral infectivity) ranging from 0.01 to 317 ng/ml (Appendix Table S2). These antibodies used different VH and VL genes, with a slight bias toward VH3 and VH4, carried heavy chain complementarity-determining region 3 (H-CDR3) of different lengths (11–21), and had a variable load of somatic mutations (Appendix Table S2). HRIG and three other human monoclonal antibodies in clinical development (CR57, CR4098, and RAB1) were used as a reference. As expected, all antibodies bound to the CVS-11 RABV G protein by ELISA.

In order to understand whether the cognate epitope is conformational or not, RABV G protein was run on a SDS–PAGE gel under reducing or non-reducing conditions and probed by Western blot with all the isolated human monoclonal antibodies. With a few exceptions (RVB143, RVC44, and RVC68), all antibodies did not bind to RABV G protein under reducing conditions, thus suggesting that the epitopes recognized, in most cases, are conformational (Appendix Table S2).

### Antibody competition studies: determination of antigenic sites on RABV G protein

Competition studies were then performed to determine the spatial proximity of each of the conformational epitopes recognized by the selected neutralizing monoclonal antibodies. The two reference antibodies CR57 and CR4098 were previously shown to recognize G protein antigenic sites I and III, respectively (Marissen *et al*, 2005; de Kruif *et al*, 2007), and were therefore used in this assay as probes to map the specificity of the other antibodies. Results shown in Fig 2 were used to cluster the 21 tested antibodies into 6 groups.

RVA125, RVC3, RVC20, and RVD74 antibodies were assigned to the antigenic site I group, according to the competition with CR57 and their reciprocal competitions. Interestingly, the binding of antigenic site I antibodies to G protein enhanced the binding by several non-antigenic site I antibodies. RVA122, RVA144, RVB492, RVC4, RVC69, RVC38, and RVC58 were assigned to the antigenic site III group, according to the competition with CR4098 and their reciprocal competitions. RVC58 showed only a partial competition with CR4098 (64%) and with antibodies that bind to sites different from site III, or I, suggesting that the RVC58 antibody recognizes a yet undefined epitope that only partially overlaps with the one recognized by CR4098. The binding of RVB181, RVC56, RVB185, RVC21, RVB161, and RVC111 was blocked by antigenic site III antibodies, but reciprocally, these antibodies did not block binding of several other antigenic site III antibodies, such as CR4098, RVC4, and RVC69. In interpreting competition results, it should be taken into account that when two epitopes overlap, or even when the areas covered by the arms of the two antibodies overlap, competition should be almost complete and mutually cross-competitive. Thus, only marked mutual cross-competition should be taken as unequivocal evidence of overlapping epitopes, since weak or one-way inhibition may simply reflect a decreased affinity due to steric or allosteric effects. Thus, the latter results suggest that these antibodies form a third cluster that recognizes a distinct, hereafter

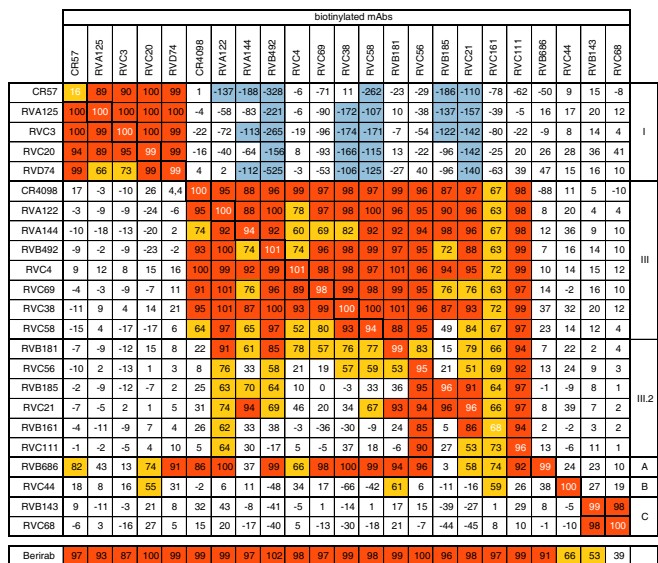

**Figure 2. RABV G protein antigenic site mapping using cross-competition ELISA-based binding studies.**
Monoclonal antibody cross-competition matrix performed by ELISA on the 21 isolated antibodies and two reference antibodies of known epitope specificity (CR57 and CR4098). The percentage of binding inhibition of the biotinylated antibodies (upper row) by the unlabeled antibodies listed in the left column is shown. Results are classified using color shading codes with values ≥ 80% in orange, < 80% and ≥ 50% in yellow, < −100% in light blue, and no shading for values < 50%.

dubbed III.2, antigenic site. Three additional sites were further defined and named A, B, and C. Site A is defined by the unique antibody RVB686, whose binding compromises the binding of the majority of the labeled antibodies in the panel, but reciprocally the binding of the labeled RBV686 is not blocked by any antibody in the panel. These results might suggest that RVB686 binding induces an allosteric effect on the G protein that compromises the binding of most other antibodies. Site B is defined by antibody RVC44, whose binding is not blocked by any other antibody in the panel. Similarly, site C is defined by antibodies RVB143 and RVC68, which also recognize a unique and distinct site as compared to all the other antibodies.

### Identification of broad-spectrum lyssavirus-neutralizing antibodies

Twelve of the 21 antibodies were selected for testing based on their neutralizing potency and recognition of distinct sites on the RABV G protein. In addition, CR57, CR4098, RAB1, and Berirab® (HRIG) were included for testing against a large panel of lyssaviruses using pseudotyped ($N = 22$) and infectious viruses ($N = 16$) covering RABV, LBV, MOKV, DUVV, EBLV-1, EBLV-2, ABLV, IRKV, KHUV, ARAV, SHIBV, BBLV, IKOV, and WCBV species (Fig 3A and B) (all viruses that neutralized with an $IC_{50}$ or $IC_{90} < 10,000$ ng/ml were scored as positive).

Among the antigenic site I antibodies tested in the pseudotyped neutralization assay (Wright *et al*, 2008, 2009), RVC20 showed the largest breadth of reactivity being able to neutralize all phylogroup I viruses tested as well as SHIBV from phylogroup II and IKOV from

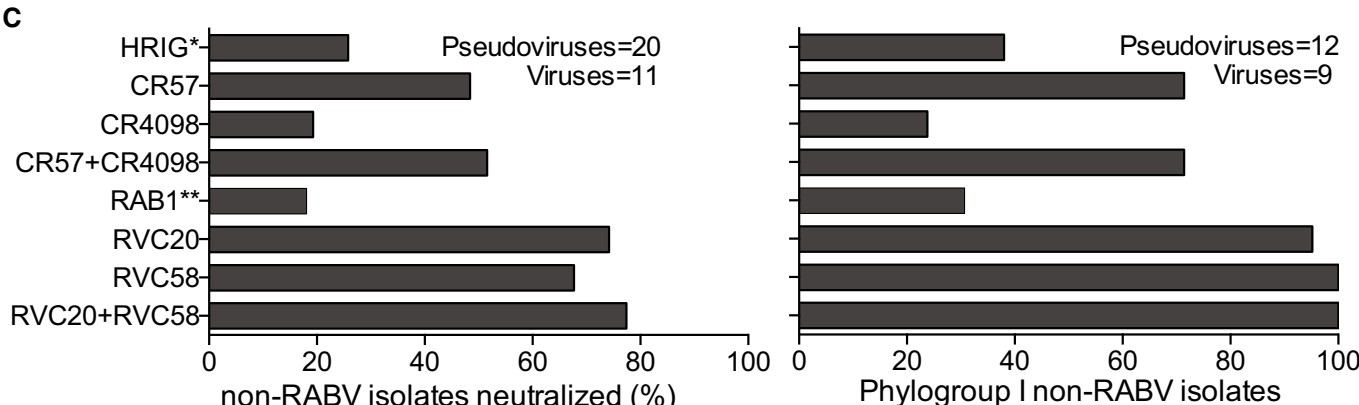

Figure 3.

**Figure 3.  Neutralization of lyssaviruses by human monoclonal antibodies.**

A selection of 12 human monoclonal antibodies, three reference antibodies (CR57, CR4098, and RAB1) and the polyclonal human immunoglobulins (HRIG, Berirab®) were tested for their neutralizing activity against 13 different lyssavirus species using pseudotyped viruses or live viruses. Complete viral strain designations are shown in Appendix Table S4.

A  Results of neutralization assays using 22 pseudotyped viruses expressed as inhibitory concentration 90 ($IC_{90}$).

B  Results of neutralization assays using 15 live viruses expressed as inhibitory concentration 50 ($IC_{50}$).

C  Summary of the percentage of non-RABV lyssavirus isolates and phylogroup I non-RABV lyssavirus isolates neutralized with $IC_{50}$ (for viruses) or $IC_{90}$ (for pseudotyped viruses) below 10,000 ng/ml for RVC20, RVC58, CR57, CR4098, and RAB1 monoclonal antibodies, HRIG or a combination of RVC20 with RVC58 or CR57 with CR4098. Color coding indicates the potency, with $IC_{90}$ (for pseudotyped viruses) or $IC_{50}$ (for viruses) < 100 ng/ml in red shading, 100 ng/ml < $IC_{50}$ < 1,000 ng/ml in orange shading, and $IC_{50} \geq 1,000$ ng/ml in yellow shading. $IC_{50}$ > 10,000 ng/ml were scored as negative. *HRIG was scored as negative when $IC_{50}$ or $IC_{90}$ was > 100,000 ng/ml; **RAB1 was tested against 20 pseudotyped viruses and 9 viruses.

putative phylogroup IV (Fig 3A). As a comparison, the antigenic site I antibody CR57 did not neutralize EBLV-1, SHIBV, and IKOV isolates. When tested on infectious viruses (Cliquet *et al*, 1998; Warrell *et al*, 2008), RVC20 broadly neutralized most of the RABV, DUVV, EBLV-1, EBLV-2, ABLV, and BBLV isolates tested as well as the phylogroup II MOKV (Fig 3B). In the same analysis, CR57 did not neutralize EBLV-1 and MOKV isolates.

Among the antigenic site III antibodies tested in the pseudotyped neutralization assay, RVC58 potently neutralized all phylogroup I viruses with an $IC_{90}$ of < 10 ng/ml. As a comparison, the antigenic site III antibodies CR4098 and RAB1 were less broad and potent and were unable to neutralize most of the non-RABV tested. When tested on infectious viruses, RVC58 potently neutralized all phylogroup I viruses tested. In the same analysis, CR4098 and RAB1 showed a limited breadth of neutralization. Of note, antibody RVC68 neutralized all phylogroups I and II pseudotyped viruses tested (only WCBV was not neutralized), although with $IC_{90}$ values 10- to 100-fold higher than compared to RVC20 and RVC58 (Fig 3A). When tested on infectious viruses, antibody RVC68 was, however, not able to effectively (i.e. $IC_{50}$ < 10,000 ng/ml) neutralize all phylogroups I and II isolates tested (Fig 3B).

Limiting the analysis of antibody breath to non-RABV lyssaviruses, RVC58 (antigenic site III) was able to neutralize 68% of all non-RABV lyssaviruses tested and, remarkably, all the phylogroup I non-RABV lyssaviruses tested (Fig 3C). In comparison, antibody CR4098 and RAB1 neutralized only 19 and 18% of the non-RABV lyssaviruses and 24 and 31% of phylogroup I non-RABV lyssaviruses, respectively. Further analysis showed that RVC20 (antigenic site I) was able to neutralize 74 and 95% of the non-RABV lyssaviruses and phylogroup I non-RABV lyssaviruses, respectively. In contrast, antibody CR57 only neutralized 48 and 71% of the non-RABV lyssaviruses and phylogroup I non-RABV lyssaviruses, respectively. When combined, RVC58 and RVC20 covered 77 and 100% of the non-RABV lyssaviruses and phylogroup I non-RABV lyssaviruses, respectively, while CR57 and CR4098 covered 52 and 71% of the non-RABV lyssaviruses and phylogroup I non-RABV lyssaviruses, respectively. HRIG was also tested against the same panel of pseudotyped and wild-type viruses and covered 26 and 38% of the non-RABV lyssaviruses and phylogroup I non-RABV lyssaviruses.

The analysis of the neutralizing activity of antibodies RVC20 and RVC58 and of the reference antibodies CR57, CR4098, and RAB1 was then extended to a large panel of RABV isolates ($n = 35$, 26 viruses and 9 pseudotyped viruses), which are representative of all circulating lineages (i.e. American, Asian, Cosmopolitan, Africa 2, Africa 3, and Arctic/Arctic-like lineages) (Fig 4 and Appendix

Table S3). All 35 RABV isolates were effectively neutralized by RVC20 and RVC58 antibodies with $IC_{50}$ or $IC_{90}$ values ranging from 0.1 to 140 ng/ml. As a comparison, CR57, CR4098, and RAB1 neutralized all the RABV tested but with a significantly lower potency. Similar to RVC20 and RVC58, HRIG neutralized the large majority of RABV strains tested with a narrow range of $IC_{50}$ ranging from 1,000 to 10,000 ng/ml. Importantly, CR4098 and RAB1 showed a broader range of IC50/IC90 values (0.7–23,600 ng/ml, 1–4,153 ng/ml, respectively), neutralizing six and three RABV isolates, respectively, with $IC_{50}$ > 1,000 ng/ml, a concentration which is likely not to be effective in PEP. This analysis was extended to an additional 7 RABV isolates for which we tested the ability of the antibodies to bind to G protein-transfected cells by flow cytometry (Appendix Table S3). All these RABV strains were recognized by RVC20 and RVC58 (CR57 did not bind to the 09029NEP and to the RV/R3.PHL/2008/TRa-065 and RAB1 to the 91001USA strain), thus extending the number of isolates analyzed to 42.

### Epitope mapping using antigenic site swapping in chimeric pseudoviruses

In order to better define the epitope specificity of the 12 selected human monoclonal antibodies, they were tested against chimeric and mutant RABV and LBV pseudotyped viruses. In particular, the amino acid changes K226E, K226N, G229E, and N336D found in the previously described CR57 and CR4098 viral escape mutants (Bakker *et al*, 2005; Marissen *et al*, 2005) were introduced into the CVS-11G gene and the corresponding mutant pseudotyped viruses produced. In addition, chimeric CVS-11 pseudotyped viruses, called C1L and C3L, were generated, in which the antigenic sites I and III residues of CVS-11 were replaced by the corresponding residues from LBV (strain NIG56-RV1) (Fig 5A). Conversely, the antigenic sites I and III residues of LBV were replaced by the corresponding residues from CVS-11 to generate the chimeric LBV pseudotyped viruses L1C and L3C, respectively. A similar approach was used for antigenic sites IIa, IIb, and IV to produce the pseudotyped viruses, L2aC, L2bC, C2bL, C2a2bL, L2a2bC, C4L, and L4C. In the case of antigenic site "a", the KG motif is conserved in LBV and so two alanine residues were introduced in the CVS-11 virus to generate the mutant, called LaA.

The panel of 12 selected antibodies as well as the reference antibodies CR57 and CR4098 were tested at 15 μg/ml for their ability to neutralize the 19 mutant pseudotyped viruses and compared with the corresponding parental CVS-11 and LBV. The results of this analysis are summarized in Fig 5B. The neutralizing activity of CR57, RVC20, and RVC3 against CVS-11 is abolished when the

antigenic site I from LBV is swapped into CVS-11 (mutant C1L). While LBV was not neutralized by these antibodies, the chimeric LBV pseudotyped virus L1C carrying the CVS-11 antigenic site I was neutralized by CR57, RVC20, and RVC3. These results are in agreement with the competition results shown in Fig 2 and confirm that RVC20 and RVC3 recognize an epitope in the antigenic site I of RABV G protein. When CR4098 and all the other remaining antibodies in the panel were tested against C1L and L1C chimeric pseudotyped viruses, their neutralizing activity was not altered compared to the parental CVS-11 and LBV viruses, thus confirming that the antigenic site I is not part of their epitope. Finally, CR57 and RVC20 antibodies (but not RVC3) were unable to neutralize the CR57 CVS-11 escape mutants K226E, K226N, and G229E. These results indicate that RVC3 recognizes an epitope in antigenic site I which is distinct from that recognized by CR57 and that RVC20 recognizes an epitope overlapping with that recognized by CR57. However, the finding that RVC20 has a broader reactivity against non-RABV lyssaviruses (Fig 3) compared to CR57 indicates that the RVC20 antibody may recognize a more conserved epitope.

Similarly, the neutralizing activity of CR4098, RVA122, RVA144, RVB185, RVC21, RVC58, and RVC111 against CVS-11 is abolished when antigenic site III from LBV is swapped into CVS-11 (mutant C3L). While LBV was not neutralized by the antigenic site III and III.2 antibodies, the chimeric LBV pseudotyped virus L3C carrying the CVS-11 antigenic site III was neutralized efficiently by CR4098, RVA144, RVB185, and RVC21. These results indicate that antibodies RVA144, RVB185, and RVC21 recognize an epitope which is similar to that recognized by CR4098. In the case of the RVC38 antibody, the swapping of antigenic site III from LBV into CVS-11 (C3L) did not affect its neutralizing activity but, similar to the other antigenic site III antibodies described above, the L3C pseudotyped virus is neutralized by RVC38. This finding suggests that the recognition of antigenic site III residues by the RVC38 antibody is sufficient to neutralize the chimeric LBV but also that its epitope in RABV CVS-11 is formed by additional residues surrounding antigenic site III. Finally, in the case of RVA122, RVC58, and RVC111 antibodies, the swapping of antigenic site III residues into LBV resulted in partial or complete loss of neutralizing activity, thus

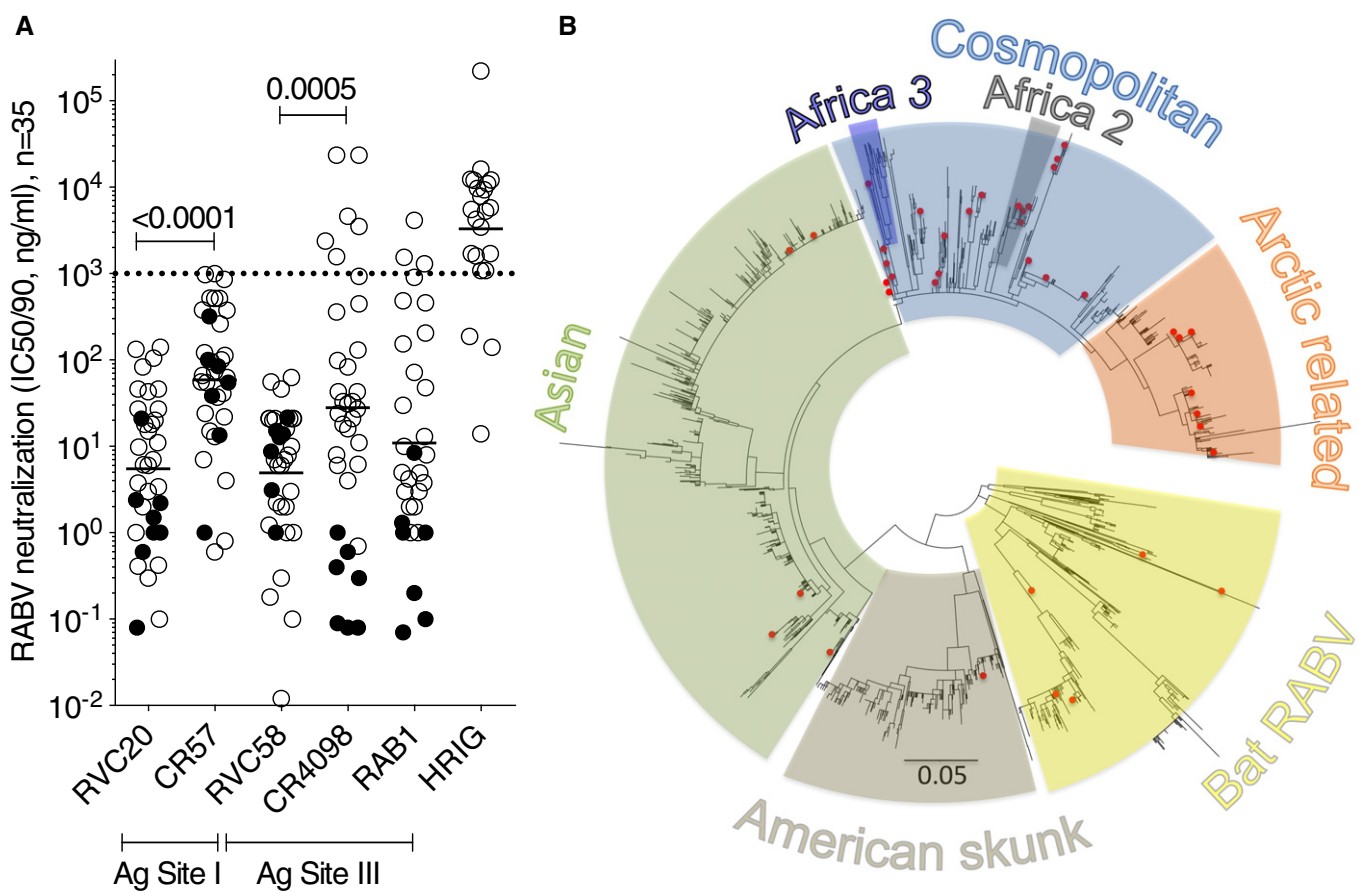

**Figure 4. RVC20 and RVC58 potently neutralize multi-lineage RABV isolates.**

A  Neutralization of RABV isolates by the selected RVC20 and RVC58 antibodies from our panel, the reference CR57, CR4098, and RAB1 antibodies and HRIG. Tested using pseudotyped viruses (filled circles, $n = 9$; shown are $IC_{90}$ values) or live viruses (empty circles, $n = 26$; shown are the $IC_{50}$ values). The dotted line indicates a threshold for neutralization above 1,000 ng/ml. The geometric mean value for each data set is shown. The P-values of Wilcoxon matched-pairs signed-rank tests are shown.

B  Phylogenetic tree of 2,215G protein sequences retrieved from public databases. Highlighted with red dots are the 40 sequences of the RABV viruses tested in this work (the G protein sequence of CV9.13 strain, was not available and was therefore not included in the tree).

Source data are available online for this figure.

suggesting that their epitopes comprise additional residues to those of antigenic site III.

Finally, all antibodies, including CR4098, but with the exception of RAB1 (data not shown), RVC111 and RVC68, were able to neutralize the CR4098 CVS-11 escape mutants N336D, thus indicating that this mutation does not block the binding of these antibodies to their cognate epitopes in the context of the CVS-11G protein. These results indicate that some antibodies, such as RVA144, RVB185, and RVC21, recognize an epitope in antigenic site III that is similar to that recognized by CR4098, while others, such as RVC58, RVB492, RVC38, and RVC111 recognize distinct epitopes in the antigenic site III region. In addition, all the antibodies in our panel directed against antigenic site III, RVC58 in particular, showed a greater breadth of reactivity against non-RABV lyssaviruses as compared to CR4098 (Fig 3). The same approach did not lead to the definition of the epitopes recognized by antigenic site B and C antibodies RVC44 and RVC68.

## Analysis of the conservation of RVC20 and RVC58 epitopes within RABV isolates

The competition results shown in Fig 2 and the results of antigenic site swapping shown in Fig 5 indicate that RVC20 and RVC58 bind primarily to antigenic sites I and III, respectively. We therefore analyzed the degree of conservation of antigenic site I and III amino acid residues in 2566 sequences from independent RABV isolates retrieved from multiple public databases representative of the global RABV diversity (Fig EV2A). We found that position 231 in antigenic site I is polymorphic (Figs 6A and EV3A). RVC20 and CR57 were tested and neutralized lyssaviruses carrying leucine, serine, or proline residues at position 231 that are representative of 99.69% of the RABV analyzed (Fig EV4). Position 226 has lysine in 99.73% of the viruses and only 0.19% of viruses carry arginine. Of note, RVC20 but not CR57 neutralized most of the non-RABV phylogroup I isolates carrying arginine at position 226, thus indicating that the presence of arginine at position 226 is not always sufficient, such as in the case of CVS-11 (Fig 5), to escape RVC20 neutralization. This analysis indicates that the RVC20 antibody epitope is highly conserved in RABV. Further analysis is required to investigate the ability of RVC20 to neutralize field isolates carrying arginine at position 226. Importantly, all three CR57 and RVC20 CVS-11 escape mutants are neutralized efficiently by RVC58.

A similar analysis was performed for the antigenic site III antibody RVC58. Antigenic site III is primarily formed by residues KSVRTWNEI (consensus sequence and positions 330–338 of the

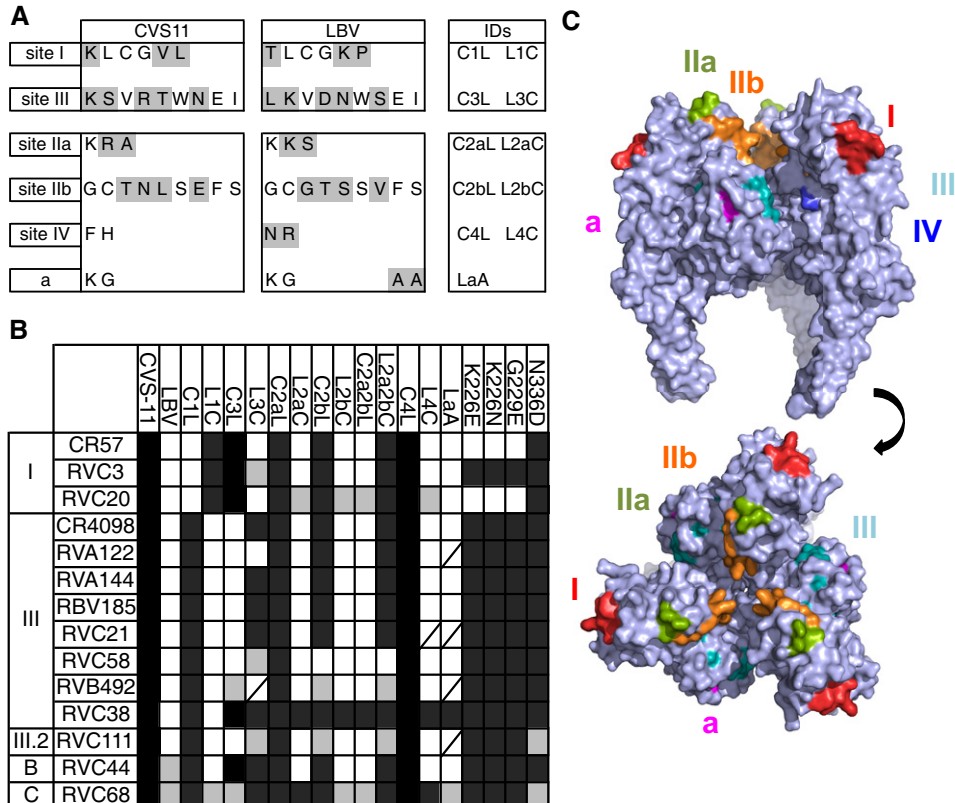

**Figure 5. Antibodies epitope mapping using antigenic site swapping in chimeric pseudotyped viruses.**

A  Sequence of the CVS-11 and LBV antigenic sites I, III, IIa, IIb, IV, and a. Highlighted in gray are the residues that differ between CVS-11 and LBV.

B  The scheme shows the results of neutralization of CVS-11, LBV (strain NIG56-RV1), different chimeric CVS-11 and LBV variants and different CVS-11 mutants by the panel of 12 selected monoclonal antibodies and the reference antibodies CR57 and CR4098. Black cells indicate full neutralization, gray cells partial neutralization, and white cells no neutralization. Strikethrough cells, not tested. Schematic showing generation of epitope swapped G protein is shown in Fig EV1.

C  Side view (upper) and top view (lower) of a surface rendering of the homotrimeric prefusion structure of VSV G (PDB, 2j6j). Rabies antigenic sites, highlighted in different colors, were superpositioned based on sequence alignment with VSV (~18% sequence identity).

RABV G protein). As described above for antigenic site I, we therefore analyzed the degree of conservation of antigenic site III amino acid residues in the panel of 2566 RABV isolates. We found that positions 330, 331, 334, 335, and 337 are highly conserved (> 99.61%), while residues 332, 333, 336, and 338 are polymorphic (Figs 6B and EV3B). RVC58 recognizes RABV and non-RABV isolates carrying multiple residues in the polymorphic positions that are representative of at least 99.80% of the RABV analyzed (Fig EV3B). In particular, 96.22% of RABV isolates have arginine at position 333. Several other residues can be found at position 333, but not aspartate, which is present in several phylogroup II viruses that are not neutralized by RVC58. Finally, RVC58 neutralized lyssaviruses carrying either asparagine, aspartate, lysine, or serine at position 336 (accounting for 99.88% of all RABV analyzed). In contrast, RABV carrying aspartate at position 336 are poorly neutralized by CR4098 and RAB1, thus indicating that approximately 4% of the circulating RABV might be resistant to CR4098 and RAB1 neutralization. Of note, the majority (59.1%) of the African RABV isolates analyzed carry a D at position 336 (Appendix Table S5). These isolates correspond to lineage Africa2. This analysis confirmed our previous neutralization results where RVC58 neutralized all phylogroup I lyssaviruses tested and indicated that the RVC58 epitope is highly conserved in RABV and non-RABV phylogroup I lyssaviruses.

### RVC58 and RVC20 antibodies protect Syrian hamsters from a lethal RABV infection

To investigate whether the antibodies RVC58 and RVC20 display neutralizing activity against a lethal RABV infection *in vivo*, we tested the two antibodies in a Syrian hamster (*Mesocricetus auratus*) model (Hanlon *et al*, 2001a). Briefly, animals (*n* = 12 per group) were challenged intramuscularly with a lethal dose of RABV CVS-11 and subsequently received post-exposure prophylaxis containing either HRIG (PEP) or an equimolar cocktail of RVC20 and RVC58 antibodies (exPEP) at different concentrations (0.045 or 0.0045 mg/kg).

Eleven out of 12 animals (92%) that were not treated after infection succumbed between day 6 and 8 (Fig 7A). The standard HRIG-based post-exposure prophylaxis (PEP) was effective in reducing the overall mortality with 8 out of 12 animals (67%) surviving the challenge. Strikingly, the combination of RVC58 + RVC20 at 0.045 mg/kg (which corresponds to 1/440 of the administered HRIG) protected 75% of the animals (9 of 12) while a 10 times lower dose of RVC58 and RVC20 (0.0045 mg/kg) protected only 33% of the animals (4/12). This suggests that 0.045 mg/kg RVC58 + RVC20 is equivalent to the 20 mg/kg HRIG dose.

To determine the effect of the antibodies mixture on vaccine potency, an *in vivo* experiment was performed in hamsters. Briefly, vaccine immunogenicity was assessed through serological testing, in the presence of HRIG or RVC20 and RVC58 cocktail administration. The results demonstrated that both HRIG (20 mg/kg) and RVC58 + RVC20 (0.045 mg/kg) did not reduce the endogenous hamster IgG-binding antibody response to the RABV G protein (Fig 7B) as compared to animals receiving vaccine alone. Of note, the level of neutralizing antibodies in animals administered with antibody cocktail (both 0.045 and 40 mg/kg) is comparable to that elicited by the vaccine alone or by the standard PEP (vaccine and HRIG) and in most animals, the neutralizing titer is above 10 IU/ml

and never below the threshold of 0.5 IU/ml (Fig 7C). Finally, while high levels of human antibodies (above 10 μg/ml) were found on day 44 in animals treated with 20 mg/kg of HRIG or 40 mg/kg of RVC58 + RVC20, undetectable or low levels of human IgG were found in the sera of animals treated with 0.045 mg/kg of RVC58 + RVC20 (Fig 7D). These results suggest that a dose of 0.045 mg/kg RVC58 + RVC20, which was shown to be protective in PEP, does not compromise the production of virus-neutralizing antibodies elicited in animals upon RABV vaccination.

## Discussion

Since they were first generated in 1975 (Kohler & Milstein, 2005), using the hybridoma technology, monoclonal antibodies have been instrumental for a wide range of applications in research, diagnosis and therapy of cancer, as well as in inflammatory and infectious diseases. In this study, we interrogated the memory B-cell repertoire of four RABV vaccines that had been pre-selected for the presence of serum antibodies capable of broadly neutralizing multiple lyssavirus species. The isolation of monoclonal antibodies from human B cells has already proven successful in the identification of several broadly neutralizing antiviral antibodies (Corti & Lanzavecchia, 2013). These could be used as probes to identify unique epitopes for the design of new vaccines capable of conferring a broad protection, but also for the development of more effective and convenient antigen-based diagnostic assays. The analysis of the specificity of the panel of human neutralizing antibodies isolated in this study unveiled a complex antigenicity of the lyssavirus glycoprotein, with new epitopes likely involved in eliciting protective host immune response. In addition to the two monoclonal antibodies selected for *in vivo* studies (i.e. RVC20 and RVC58), we have identified several others of interest, whose specificity and properties will require further investigations. Of note, one of these antibodies, namely RVC68, showed an extraordinary breadth of reactivity across phylogroups I and II lyssaviruses and recognized a linear epitope yet to be determined.

Behring and Kitasato pioneered the use of passive antibody therapy in the early 1890s when they showed that this approach could protect against diphtheria and tetanus (Kitasato, 1890). Although therapy based on animal sera was shown to be effective for diphtheria and other infectious diseases, their use was associated with hypersensitivity reactions and serum sickness caused by large amounts of animal proteins. For this reason, in several cases, such as for cytomegalovirus, varicella zoster virus, hepatitis B virus, and respiratory syncytial virus, the development of human hyperimmune immunoglobulin preparations was preferred.

In the case of rabies, several animal studies in the 1930s provided evidence that anti-rabies virus serum increased the incubation period and contributed to survival (Babes & Lepp, 1889; Habel, 1954), and subsequent studies showed that anti-rabies virus serum combined with vaccination was more efficient than vaccination or serum alone (Koprowski *et al*, 1950). To determine whether a combination of vaccine and serum would generate similar results in humans, the WHO Expert Committee on Rabies assessed a series of studies for the role of serum and vaccination. The most important study was performed in Iran in 1954 on 29 individuals bitten by the same rabid wolf, demonstrating that the combination of

    

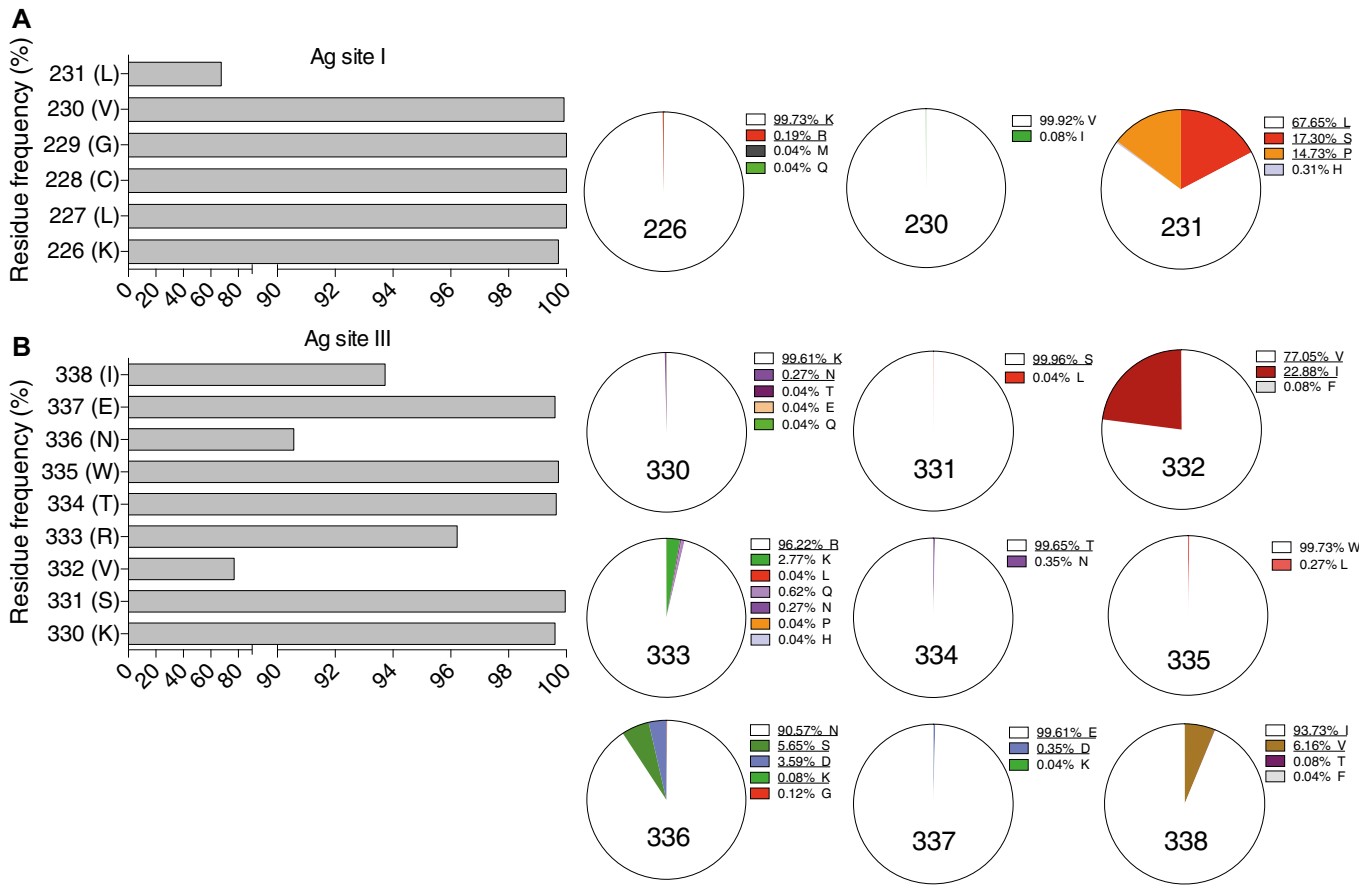

**Figure 6.  RVC20 and RVC58 target highly conserved epitopes in antigenic sites I and III.**
Level of amino acid residue conservation in antigenic sites I and III as calculated by the analysis of the G protein sequences from 2,566 RABVs. Pie charts show the detailed distribution of amino acid usage at each position. Underlined residues indicate that viruses carrying the corresponding residue in that position are neutralized by either RVC20 or RVC58.

A   Frequency of amino acid residues in antigenic site I.
B   Frequency of amino acid residues in antigenic site III.

Source data are available online for this figure.

serum and vaccine given within 2 days after exposure was clearly more effective than vaccine alone (Baltazard & Bahmanyar, 1955; Habel, 1957). Several subsequent studies in uninfected individuals showed that co-administration of serum antibodies and vaccine reduced the active endogenous humoral response (Atanasiu *et al*, 1967); however, booster doses of vaccine partly overcame this interference (Wiktor *et al*, 1971). The mechanism of RIG action is based on passive antibody administration, which can confer immediate protection through the neutralization of rabies virus at the site of infection, unlike vaccination where the stimulation of protective immunity is delayed. However, an active immune response stimulated by the vaccine can then be developed in the absence of spread of the virus to the CNS. Although no definitive protective titer is defined for all possible exposure scenarios, the achievement of a serum-neutralizing antibody titer equal to or > 0.5 IU/ml is considered the protective threshold to be achieved at day 14 after the beginning of PEP and represents the endpoint of ongoing Phase 1 and 2 trials with monoclonal antibody-based PEP (Bakker *et al*, 2008; Manning *et al*, 2008). In a previous comparative study, no

significant differences in neutralizing titers elicited by vaccination were observed when either the CR57 + CR4098 mixture or the HRIG were administered according to a PEP protocol (Goudsmit *et al*, 2006). Similarly, the RVC20 + RVC58 antibody mixture (0.045 mg/kg) did not interfere with vaccination response in hamsters. Additionally, we found that the neutralizing antibodies detected in the peripheral blood of hamsters more than 40 days after administration of either HRIG or RVC20 + RVC58 (0.045 mg/kg), were mainly hamster antibodies derived from the endogenous immune response (Fig 7B and D). Of note, our PEP antibody cocktail (0.045 mg/kg) had almost been fully cleared by the organism about 40 days after administration, conversely to what happens when a higher dose of immunoglobulins [either HRIG (20 mg/kg) or RVC20 + RVC58 (40 mg/kg)] is used. As for the RVC20 + RVC58 (40 mg/kg) antibody mixture, although the endogenous response elicited indicated that an interference between monoclonal antibodies and vaccine had somehow occurred, hamsters still had a high neutralizing titer in peripheral blood over 40 days after administration and were therefore still potentially protected against a lethal

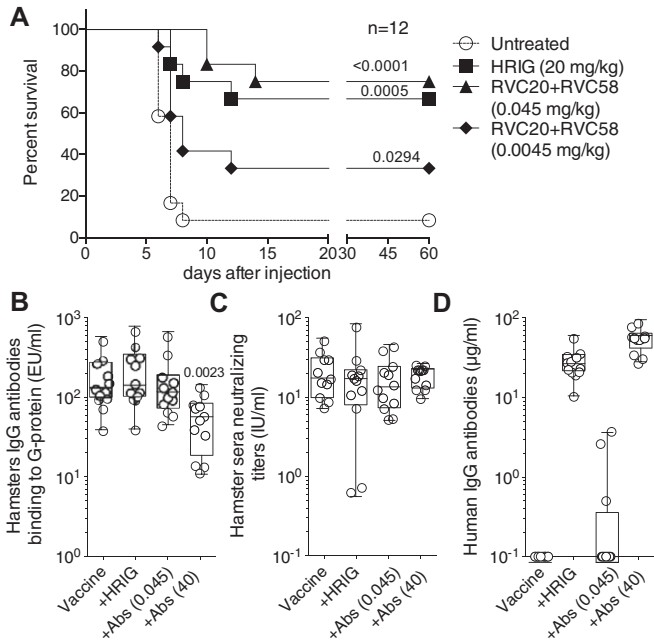

**Figure 7. RVC20 and RVC58 antibodies protect Syrian hamsters from lethal RABV infection.**

A   Percent survival in Syrian hamsters infected with RABV CVS-11 isolate and then left untreated or treated with the standard PEP (HRIG and vaccination) or with two experimental PEP protocols replacing HRIG with different doses of a cocktail of RVC20 and RVC58 monoclonal antibodies (and vaccination). Kaplan–Meier survival curves are shown by plotting percent survival against time (in days). Mantel–Cox test performed to compare treated groups versus untreated group.

B   Titers of hamster IgG antibodies binding to G protein measured in sera collected 42 days after immunization. The P-value of a Mann–Whitney test comparing the vaccine alone group with all others is shown.

C   RABV-neutralizing antibodies measured in sera collected 42 days after immunization.

D   Residuals human IgG antibodies measured in sera collected 42 days after immunization.

Data information: Box and whiskers plot (B-C): box containing 50% with median and whiskers minimum to maximum values.

Source data are available online for this figure.

RABV challenge. This finding certainly deserves further investigations, as it has the potential to break the paradigm on which post-exposure prophylaxis approaches are based (i.e. immediate administration of RIG could be safely delayed, but only of one week, if vaccine is timely administered).

The supply of HRIG is dwindling due to the difficulty in finding human donors and its expense (approximately USD 600–1200 per adult treatment). Indeed, only about 1 million doses of RIG (0.7 million ERIG and 0.3 million HRIG; excluding the Chinese domestic market) are produced and sold every year and about 60% of the people with category III exposures do not receive RIGs (Bourhy et al, 2009). ERIG production is also difficult due to the disappearance of many local producers and ethical issues (the production has been condemned by animal protection groups). The lower safety derived from the use of ERIG was compensated by the development of F(ab')2 ERIG (purified pepsin-digested ERIG, Favirab). However, the reduced half-life of F(ab')2 products might

have contributed to a few anecdotal PEP failures (Ertl, 2009) and related data derived from animal studies has shown that intact immunoglobulin products are more effective for rabies PEP than those comprising F(ab')2 fragments (Hanlon et al, 2001b). Another general drawback of RIGs is that most of the virus-specific antibodies are non-neutralizing and only a small proportion of the many antibodies are pathogen specific.

The transition from RIG to monoclonal antibody-based PEP was therefore strongly recommended by the WHO with the aim to achieve an adequate supply, a reduction in the production costs, a reduction in adverse reaction risks, and the availability of consistently active batches. In addition, since monoclonal antibodies come in the form of a concentrated product, they can be more effective than RIG at wound infiltration and therefore reduce the introduction of excess volume at the site of intramuscular injection. Another advantage may derive from formulation studies to develop monoclonal antibodies in highly stable formats (e.g. lyophilized) that would allow long-term storage as well as more convenient supply to rural areas. According to WHO recommendations, the best approach to replace RIG with monoclonal antibodies is based on the development of a cocktail of antibodies able to reduce the risk of PEP failure. The best characteristics for a cocktail of RABV-neutralizing antibodies are as follows: (i) high potency, (ii) recognition of distinct non-overlapping antigenic sites, and (iii) high breadth of reactivity for complete coverage of field RABV isolates. Another important feature is that the activity of the antibodies forming the cocktail should not rely on a synergistic effect. Indeed, although synergy may result in potent neutralization of a virus harboring all epitopes of the individual cocktail antibodies, this potency could be significantly reduced for a virus in which any of these epitopes is not available. Importantly, the RVC20 and RVC58 antibodies did not show any synergistic neutralizing activity in vitro (Appendix Figure S1).

In this study, we selected two human monoclonal antibodies (RVC58 and RVC20) from vaccinees for their ability to bind to two distinct antigenic sites (sites I and III) on the RABV G protein. In addition to this, they were able to potently neutralize each RABV isolate in our panel, representing all lineages, and all phylogroup I non-RABV isolates. Indeed, the identification of rare, broadly reactive antibodies such as those selected in this study can increase the barrier to the occurrence of resistance since they can cope with a higher degree of variability in their cognate epitopes. In this regard, it is important to note that viral escape mutants might have an impaired in vivo fitness and this might particularly be the case for viral escape from multiple, broadly reactive antibodies. An additional important feature of RVC20 and RVC58, similar to HRIG, is their ability to neutralize all the RABV tested within a narrow and similar range of antibody concentrations, in contrast to CR57, CR4098, and RAB1. In previous studies, it was found that the antigenic site I CR57 antibody was not able to neutralize 2 out of 26 viruses tested and that from the analysis of a database of 229 isolates ~1% contained mutations that would most likely abrogate binding of CR57 (Bakker et al, 2005; Marissen et al, 2005). In the case of the antigenic site III CR4098 antibody, it was found that in a database of 123 RABV isolates 5 out of 123 (4%) harbored the N336D mutation (Bakker et al, 2005). While none of the RABV isolates carried mutations in both the CR57 and CR4098 epitopes, the existence of naturally occurring strains resistant to one of the

two antibodies in the cocktail (1–4%, with higher frequency in endemic areas in Africa, which has the second highest number of cases after Asia) poses a higher risk of *in vivo* selection of mutants able to escape from the second antibody, thus leading to a potential risk of PEP failure. In the case of the RAB1 antibody, currently being tested in a Phase 2/3 trial in India as a single antibody, two out of 25 isolates tested were not neutralized and three were poorly neutralized (Sloan *et al*, 2007). In this case the risk of PEP failure is, at least in principle, higher than in the case of the CR57 and CR4098 antibody cocktail. These and our results suggest that CR4098 and RAB1 have a limited breadth of reactivity toward non-RABV isolates and that a significant fraction of the RABV isolates tested are not or only poorly neutralized by these antigenic site III antibodies. It is also important to note that the analysis of different HRIG preparations revealed that not all RABV are covered. In particular, a comparison of Imogam HRIG (Sanofi) and BayRab (Bayer) showed that one RABV strain of bat origin was neutralized by BayRab HRIG, but not by the Imogram HRIG (Goudsmit *et al*, 2006). Our results, as well as results from other studies (Brookes *et al*, 2005; Hanlon *et al*, 2005; Horton *et al*, 2010), showed that HRIG has a limited efficacy toward non-RABV isolates. Another advantage of replacing HRIGs with a cocktail of two broadly neutralizing antibodies is represented by the possibility to use them in the PEP of phylogroup I non-RABV isolates. Finally, the high *in vitro* potency of RVC20 and RVC58 antibodies and the *in vivo* results presented in this study demonstrated that only a limited amount of these antibodies is needed to protect from lethal infection (equivalent to ~3 mg for a 70 kg adult). Considering the marked reduction in antibody production costs (Kelley, 2009), this amount would be compatible with a considerably lower price (e.g., 1–10 US dollar) for GMP-grade antibodies as compared to that of HRIGs.

In conclusion, the combination of the RVC20 and RVC58 antibodies represents a treatment with an unprecedented breadth and potency for the development of a low-risk and affordable product to replace RIGs in rabies PEP.

# Materials and Methods

### Isolation of monoclonal antibodies

IgG$^+$ memory B cells were isolated from cryopreserved PBMC using CD22 microbeads (Miltenyi Biotec), followed by depletion of cells carrying IgM, IgD, and IgA by cell sorting, and immortalized with EBV and CpG in multiple replicate wells as previously described (Traggiai *et al*, 2004). Culture supernatants were tested for their ability to neutralize CVS-11 RABV pseudotyped virus infection in a micro-neutralization assay. Positive cultures were collected and expanded. From positive cultures, the VH and VL sequences were retrieved by RT–PCR. RVC20 and RVC58 antibodies were cloned into human IgG1 and Ig kappa or Ig lambda expression vectors (kindly provided by Michel Nussenzweig, Rockefeller University, New York, NY, USA) essentially as described (Tiller *et al*, 2008). Monoclonal antibodies were produced from EBV-immortalized B cells or by transient transfection of 293 Freestyle cells (Invitrogen). Supernatants from B cells or transfected cells were collected and IgG were affinity purified by Protein A or Protein G chromatography (GE Healthcare) and desalted against PBS.

### Production of pseudotyped viruses and neutralization assay

Human embryonic kidney 293T clone 17 cells (HEK 293T/17; ATCC CRL-11268) were used for production of the lentiviral pseudotypes. Neutralization assays were undertaken on BHK-21 cells clone 13 (ATCC CCL-10). In a 384-well plate, pseudotyped virus that resulted in an output of $50–100 \times 10^4$ relative light units (RLU) was incubated with doubling dilutions of sera or antibodies for 1 h at 37% (5% $CO_2$) before the addition of 3,000 BHK-21 cells. These were incubated for a further 48 h, after which supernatant was removed and 15 μl SteadyLite reagent (Perkin Elmer) was added. Luciferase activity was detected 5 min later by reading the plates on a Synergy microplate luminometer (BioTek) (Wright *et al*, 2008). The reduction of infectivity was determined by comparing the RLU in the presence and absence of antibodies and expressed as percentage of neutralization. The neutralization potency for the monoclonal antibodies is here measured as $IC_{90}$, which was defined as the antibody concentration at which RLU were reduced 90% compared with virus control wells after subtraction of background RLU in cell control wells (ID50 for the sera, that is the dilution of sera at which RLU were reduced 50%). $ID_{50}$ values for the sera correspond to the dilution at which RLU were reduced 50%. Antigenic site swapping between the CVS-11 (accession no. EU352767) and LBV.NIG56-RV1 (accession no. EF547431) G genes was undertaken using overlapping PCR (Heckman & Pease, 2007) and confirmed by sequence analysis. The resulting G genes were subsequently used to generate pseudotyped viruses and titrated on BHK cells to ensure the mutations did not affect the binding and entry function of the G proteins.

### Binding assay

A standard ELISA was used to determine binding of serum antibodies or monoclonal antibodies to RABV G protein (CVS-11). Briefly, ELISA plates were coated with RABV G protein at 5 μg/ml, blocked with 10% FCS in PBS, incubated with sera or human antibodies, and washed. Bound antibodies were detected by incubation with AP-conjugated goat anti-human IgG (Southern Biotech). Plates were then washed, substrate (p-NPP, Sigma) was added and plates were read at 405 nm. The relative affinities of sera binding or monoclonal antibody binding were determined by measuring the dilution of sera (ED50) or the concentration of antibody (EC50) required to achieve 50% maximal binding at saturation.

### Western blot analysis

Purified RABV G protein (prepared according to Meslin *et al*, 1996) was loaded on a 12% Tris–glycine polyacrylamide gel. Protein transfer on a PVDF membrane was performed with the iBlot blotting system from Invitrogen. PVDF membrane was blocked for 30 min with 10% non-fat dry milk in TBS–Tween. Incubation with primary antibodies against G protein was performed at 0.5 μg/ml in TBS–Tween overnight at 4°C. PVDF was washed three times with TBS–Tween and incubated for 1 h at RT with HRP-conjugated anti-human IgG antibody (GE Healthcare). PVDF membrane was washed three times with TBS–Tween and positive bands detected using ECL Plus™ Western Blotting Detection Reagent (GE Healthcare) and the LAS4000 CCD camera system.

## ELISA competition assay

CR57, CR4098, and all 21 antibodies selected were labeled with biotin and tested by ELISA in a matrix competition assay, in which unlabeled antibodies were incubated first at a concentration of 25 μg/ml on RABV G protein-coated plates, followed by the addition of a limiting concentration of biotinylated antibodies whose binding was revealed with alkaline phosphatase-conjugated streptavidin. When interpreting competition results, it should be taken into account that if two epitopes overlap, or the areas covered by the arms of the two antibodies overlap, competition should be almost complete. Weak inhibitory or enhancing effects may reflect a decrease in affinity owing to steric or allosteric effects.

## Lyssavirus cell adaptation and *in vitro* neutralization assays

Selected RABVs and non-RABV lyssaviruses were initially cultured on Neuro-2A cells (ATCC cat n. CCL-131) and further adapted on BSR cells (a clone of BHK-21). Two protocols slightly modified from fluorescent antibody virus neutralization (mFAVN) and from rapid fluorescent foci inhibition (mRFFIT) test (Cliquet *et al*, 1998; Warrell *et al*, 2008), respectively, were applied to test the potency of antibodies under study. CVS-11 working stock was amplified and titrated on either BSR or BHK-21, according to the neutralization test adopted, RFFIT or FAVN, respectively. In addition, standard FAVN and RFFIT assays were undertaken to assess the potency of tested antibodies against CVS-11. Briefly, mFAVN assays were based on standard FAVN but were undertaken on BSR cells.

## RNA extraction, RT–PCR and sequencing

Sequencing of complete G gene of the original specimens as well as of the cell-adapted lyssaviruses used as challenge viruses in the *in vitro* assays was obtained. Viral RNA was extracted using the Nucleospin RNA II kit, according to the manufacturer's instructions (Macherey–Nagel GmbH & Co., Düren, Germany). Briefly, one hundred microliters of sample suspension were used for the extraction, and RNA was eluted in a final volume of 60 μl and stored at −80°C. One-step RT–PCR amplification was performed using the Qiagen OneStep RT–PCR kit (Qiagen GmbH, Hilden, Germany) according to the manufacturer's instructions. Primers used for the amplification of complete G gene sequences are available upon request. PCR products were analyzed for purity and size by electrophoresis in 2% agarose gel after staining with GelRedTM Nucleic Acid Gel Stain (Biotium, Hayward, CA). Amplicons were subsequently purified with ExoSAP-IT (USB Corporation, Cleveland, OH) and sequenced in both directions using the Big Dye Terminator v3.1 cycle sequencing kit (Applied Biosystems, Foster City, CA, USA). The products of the sequencing reactions were cleaned up using the Performa DTR Ultra 96-well kit (Edge BioSystems, Gaithersburg, MD) and analyzed on a 16-capillary ABI PRISM 3130xl Genetic Analyzer (Applied Biosystems, Foster City, CA, USA).

## RABV sequences analysis

The occurrence of different amino acid identities at antigenic site I and antigenic site III was analyzed by downloading all RABV glycoprotein sequences present in National Center for Biotechnology Information (NCBI) Entrez Protein database (Sayers *et al*, 2009) as of November 25, 2014. The retrieved sequences were purged of those missing the complete sequence from amino acid positions 200–400 (covering Antigenic sites I and III) or containing ambiguous amino acid identities or lacking the country of origin for a total of 2,566 sequences. These sequences (including the 38 sequences of the RABV tested in our panel) were aligned using MAFFT (Katoh & Standley, 2013) to perform the amino acid distribution analysis. All algorithms are written in Java. A Multiple sequence alignment of amino acid sequences was performed on the full length 2,215 sequences (480 amino acid residues in length) using Clustal omega (Sievers & Higgins, 2014). Phylogenetic analysis of these sequences was then undertaken using the maximum likelihood method available in the PhyML package (Guindon & Gascuel, 2003). This analysis utilized the LG model of amino acid replacement with a gamma distribution of among-site rate variation.

## Ethical statement

Blood samples were collected from participants vaccinated against rabies. All donors gave written informed consent for research use of blood samples, following approval by the Cantonal Ethical Committee of Cantone Ticino, Switzerland. Animal studies were performed in strict accordance with the relevant national and local animal welfare bodies [Convention of the European Council no. 123 and National guidelines (Legislative Decrees 116/92 and 26/2014)]. The protocol was authorized by the Italian Ministry of Health (Decrees 128/2011-B and 115/2014-PR) before experiments were initiated and approved by the Committee on the Ethics of Animal Experiments of the IZSVe.

## Animal studies

All experiments were performed on female SPF Syrian hamsters (*Mesocricetus auratus*) of 6–7 weeks of age (average weight 105 g) (Charles River Laboratories). Animals were housed in individually HEPA-filtered ventilated cages, three individuals per cage, at a temperature of 22 ± 1°C, on a 12L:12D light cycle, with free access to water and food. Pressed cotton pads, mouse houses, and litter bags were used as environmental enrichment, and the standard rodent feed was weekly integrated with autoclaved sunflower seeds. In order to minimize the effect of subjective bias during allocation, animals were randomly assigned to treatment or control groups. No blinding of investigator was implemented. No samples nor animals were excluded from the analyses.

## Post-exposure prophylaxis of Syrian hamsters

Forty-eight Syrian hamsters were challenged at day 0 by the intramuscular route (gastrocnemius muscle in the right hind limb) with 0.05 ml of a 1:100 ($10^{6.76}$ $MICLD_{50}$/ml) dilution of the CVS-11 strain. Animals were given biologics or PBS (negative control) by the end of day 0. For each treated group ($n = 12$), a commercial vaccine [Rabipur®; Novartis Vaccines and Diagnostics, a purified chick embryo cell vaccine containing inactivated rabies virus (strain flury LEP), potency ≥ 2.5 International Units (IU)] was administered intramuscularly (i.m.) in the left hind limb at days 0, 3, 7, 14, and

## The paper explained

### Problem

Rabies is a neglected zoonosis mostly associated to a dog-mediated infection transmitted through a bite. Despite preventable through the application of both pre- and post-exposure prophylaxis (PEP) protocols, human deaths due to rabies are still a burden in endemic areas, with a roughly estimated 60,000 deaths each year. More than 95% of cases occur in Asia and Africa, 50% of which affect children. Current protocols of PEP include the first-aid treatment of the wound and the administration of a rabies vaccine, which is coupled with rabies immunoglobulin (RIG) administration in specific high-risk cases. More than 15 million people a year are treated after an episode of suspect or confirmed rabies exposure. However, the limited production and the high costs of RIGs make them scarcely available on the market, especially in developing countries where they are mostly needed.

### Results

We immortalized memory B cells from four vaccinated donors and then tested monoclonal antibodies for their rabies neutralizing activity and epitope specificity. We selected two antibodies, identified as RVC20 and RVC58 as highly potent and broadly neutralizing a panel of 60 lyssaviruses (including rabies and non-rabies viruses), widely representatives of the global viral diversity. The potency and breadth shown by the two selected antibodies were higher than those detected in commercially available human RIG and in previously discovered antibodies under clinical development. Our antibody cocktail (RVC20 + RVC58), experimentally administered as alternative to commercially available RIG according to current PEP protocols, was able to protect Syrian hamsters from a lethal rabies virus challenge and did not impact on vaccine immunogenicity in hamsters.

### Impact

The cocktail of antibodies proposed in our study represents a treatment with an unprecedented breadth and potency to be eventually adopted as a valid alternative to RIG in the frame of human rabies PEP. Importantly, considering the high potency of our cocktail, the amount needed in PEP would be compatible with a considerably lower price if compared to that of currently available RIG.

28 post-infection (p.i.). Hyperimmune treatment consisted of HRIG (Berirab®, 20 mg/kg, equivalent to 20 IU/kg) or candidate antibodies (0.045 mg/kg and 0.0045 mg/kg) of an equimolar mixture of RVC20 and RVC58 antibodies administered i.m. on day 0 in the right hind limb in a final volume of 50 μl. Challenged animals were observed twice a day and promptly euthanized at the onset of one of clinical signs of rabies (i.e. motoric deficit, lack of coordination, paresis, paralysis, sensory dullness). Central nervous system (CNS) tissues, namely brain, cerebellum, brainstem and spinal cord, were collected to confirm rabies virus infection by means of a standard technique, fluorescent antibody test (FAT) (OIE World Organization for Animal Health. 2013).

### Vaccine immunogenicity in non-challenged Syrian hamsters treated with the monoclonal antibodies cocktail

Forty-eight Syrian hamsters were divided in four groups ($n = 12$ per group) and vaccinated with a commercial purified chicken embryo cell vaccine (Rabipur®; Novartis Vaccines and Diagnostics, a purified chick embryo cell vaccine containing inactivated rabies virus, strain flury LEP, potency ≥ 2.5 IU). Vaccine was administered on day 0, 3, 7, 14, and 28 i.m. in the left hind limb. Three groups were concomitantly administered on day 0 with HRIG (Berirab®, 20 mg/kg) or an equimolar mixture of RVC20 and RVC58 at 0.045 or 40 mg/kg (referred as HD) that were injected i.m. in the right hind limb and in the right and left hind limbs for HD antibody cocktail administration. Sera were collected from all animals ($n = 48$) on day 44 post-vaccination (p.v.) and tested by FAVN (Cliquet et al, 1998) for the presence of rabies neutralizing antibodies and by ELISA for the presence of either G protein-specific hamster antibodies or residual human IgGs.

### Statistics

The number of individuals in each experimental group ($n = 12$ per group) was calculated using Fisher's exact conditional test for two proportions (as implemented by Proc Power twosamplefreq, SAS software) and power $1-\beta = 0.80$ ($\alpha = 0.05$). A Wilcoxon matched-pair signed-rank test was used to analyze differences in mean values between groups. Mann–Whitney and Mantel–Cox tests were also used as described in legends. $P$-values of $< 0.05$ were considered statistically significant.

**Expanded View** for this article is available online.

### Acknowledgements

This study was partially funded by the Italian Ministry of Health (RC IZSVe 08/2009). The Institute for Research in Biomedicine is supported by the Helmut Horten Foundation. AL is also supported by the Swiss Vaccine Research Institute. PDB, ERN, RA, BZ, AS, GC, and IC wish to thank Silvia Tiozzo Caenazzo and Crispina Veggiato (FAO/National RC for Rabies, IZSVe-Italy) and William G. Dundon (currently IAEA Laboratories-Austria) for their help and technical contributions. Massimo Boldrin and Franco Mutinelli (Animal Welfare Body, IZSVe) are also gratefully acknowledged for having guaranteed and substantially improved the animal welfare and care of experimental animals used in this study.

### Author contributions

PDB, IC, ALa, HB, and DC conceived the study. PDB, RW, GC, IC, FS, ALa, HB, and DC participated in the study design. AM, GA, and FV isolated the monoclonal antibodies. AM, ERN, RA, BZ, GA, FV, RL, ALe, EB, and EW characterized the monoclonal antibodies *in vitro*. AS, RL, and MF performed and analyzed viral sequencing. PDB, ERN, and RA performed the *in vivo* tests. PDB, EW, ALa, HB, and DC critically analyzed the data. PDB, EW, HB, and DC drafted the manuscript. All authors revised and accepted the manuscript in its present form.

### Conflict of interest

PDB, RA, BZ, AS, GC, and IC are employed by IZSVe. RL, ALe, and HB are employed by IP-Paris, and AM, GA, FV, and DC are employed by Humabs BioMed SA. ALa is the scientific founder of Humabs BioMed SA and holds shares in Humabs BioMed SA. This does not alter our adherence to all EMBO press journals policies on sharing data and materials.

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
