## [Review Process File · EMBO Molecular Medicine]

Development of broad-spectrum human monoclonal antibodies for rabies post-exposure prophylaxis

Paola De Benedictis, Andrea Minola, Elena Rota Nodari, Roberta Aiello, Barbara Zecchin, Angela Salomoni, Mathilde Foglierini, Gloria Agatic, Fabrizia Vanzetta, Rachel Lavenir, Anthony Lepelletier, Emma Bentley, Robin Weiss, Giovanni Cattoli, Ilaria Capua, Federica Sallusto, Edward Wright, Antonio Lanzavecchia, Hervé Bourhy, Davide Corti

Corresponding author: Davide Corti, Humabs Biomed

Review timeline:

Submission date:	02 November 2015
Editorial Decision:	18 December 2015
Revision received:	28 February 2016
Accepted:	15 February 2016

Transaction Report:

Editor: Céline Carret

1st Editorial Decision

18 December 2015

Thank you for the submission of your manuscript to EMBO Molecular Medicine. We have now heard back from two of the three referees whom we asked to evaluate your manuscript. As these two referees have very similar recommendations, we decided to go ahead and make a decision now.

Although referees do find the study suitable for publication in principle, referee 2 suggested expanding the discussion and providing additional explanations here and there.

We would welcome the submission of a revised version for further consideration and depending on the nature of the revisions, this may be sent back or not to the referees for another round of review.

In order to gain time, shall the manuscript be accepted I would also like you to address several editorial issues listed below.

I look forward to seeing a revised form of your manuscript as soon as possible and within 3 months.

***** Reviewer's comments *****

Referee #1 (Comments on Novelty/Model System):

The work is outstanding from a technical perspective. It is not particularly novel as several monoclonal antibodies have been tested previously for RIG replacement. The authors' correctly state

that the problem with MABs is lack of complete coverage against the multitude of lyssavirus. The described MABs have very broad neutralizing activity and may therefore allow for their clinical development. The animal model is appropriate.

Referee #1 (Remarks):

This manuscript, which in great detail describes the specificity of several MABs that may be suited to replace RIG, is timely and will be an important addition to our published arsenal of rabies biologics. The authors may wish to check the manuscript carefully for language - some of the sentences don't quite adhere to English rules of grammar (e.g., last two sentences of the introduction).

Referee #3 (Remarks):

The manuscript by De Benedictis describes the isolation and characterization of several monoclonal antibodies to lyssaviruses, with potential utility for rabies post-exposure prophylaxis. A total of 500 mAbs were isolated from memory B cells from vaccinees prescreened for anti-RABV activity. From these, two mAbs (RVC20 and RVC58) were shown to neutralize with greater breadth and potency than those mAbs currently under clinical development. This data is supplemented by partial epitope mapping and virological analysis. Finally, the authors show that relatively low doses of a combination of these 2 mAbs protects hamsters from a lethal RABV challenge, and assessed the effect of these mAbs on vaccine responses in hamsters. The potential superiority of these new mAbs over CR57, CR4098 and RAB1 is clearly shown here, providing the basis for future clinical studies.

Figure 2 and related text. There are some intriguing findings here. Although the reciprocal competition is very clean for mAbs to antigenic site I, it is more confusing for antigenic sites III and III.2, with mAbs RVB181, RVC56 etc. blocked by antigenic site III antibodies, but unable to block in the opposite assay. The authors speculate that these antibodies form a third cluster that recognizes a distinct site, but I am not sure this accounts for the timing. The opposite is true for RBV686. Are there potential other explanations e.g. conformational changes that could account for this?

Figure 3. the extraordinary breadth of RCV68 (despite low potency) suggests this target should be further investigated as a new conserved target. It would be worth adding this to the discussion.

Page 7. Can the authors comment on the discrepancy between neutralization of pseudotyped viruses compared to live viruses by RVC68? Is this simply a general reflection of the reduced potency of the mAb compared to others i.e. is the pseudovirus assay intrinsically more sensitive or does this reflect the use of IC50 versus IC90?

Figure 6. Why is the more detailed analysis (the pie charts) only performed for selected sites? For Ag site 1, only the 2 most polymorphic sites are described, whereas for Ag site II only one site is omitted.

Figure 7. In panel A (the challenge study) the high dose is 0.045, whereas for the vaccine responsiveness experiments in panels B-D, HD is 40mg/kg. I found this confusing and would more clearly label the axes in B-D.

Page 12. Can the authors comment on the observation that in the presence of the HD of mAbs, the binding responses are significantly lower, but neutralizing responses unaffected.

The discussion is weak. While it is important to place these data in the context of PEP, there are many other aspects (more scientific) that are not discussed at all e.g. potential new epitopes. A more thoughtful discussion would substantially strengthen this paper.

Minor:

Page 4, last line of intro - incomplete sentence

Page 7 - last paragraph. It does not appear to me that CR57 has a greater range of IC50/90 values than RVC20, though undoubtedly it is less potent

Figure 5B is extremely hard to follow - the addition of a schematic showing how the chimeras are constructed might be useful to clarify this.

1st Revision - authors' response

28 February 2016

Referee #1 (Comments on Novelty/Model System):

The work is outstanding from a technical perspective. It is not particularly novel as several monoclonal antibodies have been tested for RIG replacement. The authors' correctly state that the problem with MABs is lack of complete coverage against the multitude of lyssavirus. The described MABs have very broad neutralizing activity and may therefore allow for their clinical development. The animal model is appropriate.

Referee #1 (Remarks):

This manuscript, which in great detail describes the specificity of several MABs that may be suited to replace RIG, is timely and will be an important addition to our published arsenal of rabies biologics. The authors may wish to check the manuscript carefully for language - some of the sentences don't quite adhere to English rules of grammar (e.g., last two sentences of the introduction).

We thank the reviewer for the positive comments on our manuscript. As suggested we have checked the manuscript for language. Several sentences have been now rephrased accordingly by the English-native scientists co-authoring the study.

Referee #3 (Remarks):

The manuscript by De Benedictis describes the isolation and characterization of several monoclonal antibodies to lyssaviruses, with potential utility for rabies post-exposure prophylaxis. A total of 500 mAbs were isolated from memory B cells from vaccinees prescreened for anti-RABV activity. From these, two mAbs (RVC20 and RVC58) were shown to neutralize with greater breadth and potency than those mAbs currently under clinical development. This data is supplemented by partial epitope mapping and virological analysis. Finally, the authors show that relatively low doses of a combination of these 2 mAbs protects hamsters from a lethal RABV challenge, and assessed the effect of these mAbs on vaccine responses in hamsters. The potential superiority of these new mAbs over CR57, CR4098 and RAB1 is clearly shown here, providing the basis for future clinical studies.

We thank the reviewer for the positive comments on our study.

Figure 2 and related text. There are some intriguing findings here. Although the reciprocal

competition is very clean for mAbs to antigenic site I, it is more confusing for antigenic sites III and III.2, with mAbs RVB181, RVC56 etc. blocked by antigenic site III antibodies, but unable to block in the opposite assay. The authors speculate that these antibodies form a third cluster that recognizes a distinct site, but I am not sure this accounts for the timing. The opposite is true for RBV686. Are there potential other explanations e.g. conformational changes that could account for this?

We agree with the Referee's comments about the difficulty to define precisely antigenic sites solely on the basis of cross-competition binding studies. In interpreting competition results, it should be taken into account that when two epitopes overlap, or even when the areas covered by the arms of the two antibodies overlap, competition should be almost complete and mutually cross-competitive. Thus, only marked mutual cross-competition should be taken as unequivocal evidence of overlapping epitopes, since weak or one-way inhibition may simply reflect a decreased in affinity owing to steric or allosteric effects (see Epitope Mapping Protocols, chapter 6, Glen E. Morris, Humana Press). A more detailed definition of the epitopes of this antibody panel would require further investigation and this work could be part of a follow-up study.

Authors acknowledge that this point was not sufficiently explained in the previous version of the manuscript and have therefore addressed it accordingly (see Results page 6, lines 13-19).

Figure 3. the extraordinary breadth of RCV68 (despite low potency) suggests this target should be further investigated as a new conserved target. It would be worth adding this to the discussion.

We appreciate the Referee's point. The method used to isolate the monoclonal antibodies investigated in this study had already proven effective in identifying broadly neutralizing antiviral antibodies, which made it possible to discover conserved epitopes that may ultimately lead to design new vaccines capable of conferring broader protection (Corti and Lanzavecchia, Annual Review in Immunology 2013). In relation to rabies, the broadly neutralizing activity of the RVC68 antibody, in spite of its limited potency if compared to other antibodies isolated in the study, worth further investigation as it presumably recognizes a conserved and probably yet undetermined epitope. Possible applications of our findings may range from vaccine development, immune therapy or to the development of new diagnostic tools accounting for the wide lyssavirus diversity. Authors acknowledge that this finding was not sufficiently discussed in the previous version of the manuscript and have addressed the Referee's remark accordingly in the Discussion section (page 12, lines 15-17).

Page 7. Can the authors comment on the discrepancy between neutralization of pseudotyped viruses compared to live viruses by RVC68? Is this simply a general reflection of the reduced potency of the mAb compared to others i.e. is the pseudovirus assay intrinsically more sensitive or does this reflect the use of IC50 versus IC90?

The discrepancy in neutralisation observed between live virus and pseudotyped virus assays is likely due to more than one reason. Firstly, the density of the G protein on the surface of these viruses may differ. Data from previous studies that isolated potent and broadly neutralising influenza mAbs (Corti et al. JCI 2010) or assessed the neutralising potency of bat sera against lyssaviruses (Wright et al. Virology 2010) suggests that pseudotyped viruses have a lower density of viral envelope protein on their surface. This characteristic of PV allowed for the isolation of mAbs that bound to the HA2 stem region of influenza A viruses, which is not readily exposed on the live virus (Corti et al. JCI 2010), but also for a more accurate reflection of lyssavirus epidemiology in bats, thanks to the greater sensitivity of the PV assay (Wright et al. Virology 2010). Secondly, PV-based assays allow the study of entry inhibition however; in the case of live virus assays we also have the potential for viral replication and spread that could affect neutralisation titres. Finally, as the reviewer infers, due to the fact that the approved protocol for running each assay (PV, RFFIT and FAVN) is different this could also lead to variation in the final readout between the assays.

Figure 6. Why is the more detailed analysis (the pie charts) only performed for selected sites? For Ag site I, only the 2 most polymorphic sites are described, whereas for Ag site II only one site is omitted.

We thank the Referee for the careful review and we agree that it would be more appropriate to show the complete analysis on all positions where the degree of conservation is not equivalent to 100%. We have therefore changed **Figure 6** accordingly by adding pie charts for residues at position 230 (panel a), 330 and 335 (panel b).

Figure 7. In panel A (the challenge study) the high dose is 0.045, whereas for the vaccine responsiveness experiments in panels B-D, HD is 40 mg/kg. I found this confusing and would more clearly label the axes in B-D.

We thank the Referee for the careful review. As suggested we have labeled the axes of panels B, C and D of **Figure 7** accordingly. The amount of monoclonals (in mg/kg) used in each experimental group has been therefore indicated in parentheses.

Page 12. Can the authors comment on the observation that in the presence of the HD of mAbs, the binding responses are significantly lower, but neutralizing responses unaffected.

Similarly to the work by Goudsmith et al (2006) for the CR4098+CR57 cocktail, we also assessed the neutralizing titres detectable more than 40 days following PEP, including the administration of vaccine and the RVC20+RVC58 antibody mixture. Of note, we have further assessed whether the peripheral neutralizing titers conferring protection to hamsters may be due to either hamster endogenous post-vaccination immune response, to exogenous human antibodies due to passive immunization or by a mixture of them. We found that viral neutralization was mainly due to hamster

endogenous response when HRIG or RVC20+RVC58 (0.045mg/kg) was administered, and that our cocktail had almost been fully cleared by the organism about 40 days after administration. As for the RVC20+RVC58 (40 mg/kg) antibody mixture (dubbed HD in the previous version), although the endogenous response elicited (as detected by ELISA by measuring the levels of hamster antibodies directed against the RABV G protein) indicated that an interference between the passively administered monoclonal antibodies and vaccine had somehow occurred, hamsters still had a high neutralizing titre in peripheral blood over 40 days post administration and were therefore still potentially protected against a lethal RABV challenge. These findings merit further investigations, as it has the potential to break the paradigm on which post-exposure prophylaxis approaches are based. Authors acknowledge that this important finding was not discussed in the previous version of the manuscript and for this reason a specific comment has been now added in the discussion section (page 13, lines 10-27).

The discussion is weak. While it is important to place these data in the context of PEP, there are many other aspects (more scientific) that are not discussed at all e.g. potential new epitopes. A more thoughtful discussion would substantially strengthen this paper.

Authors acknowledge that other aspects of the study were not discussed in the previous version of the manuscript; this is why the discussion section has been strengthened with a paragraph underlining some of the most important aspects:

- (i) the potential for discovering new epitopes (i.e. that recognized by RVC68), and the relevance of identifying antigenic sites that are conserved among different lyssaviruses (pages 12, lines 6-17);
- (ii) a possible explanation of unexpected pattern of previously characterized ASIII antibodies and, more generally, of ASIII antibodies as characterized in the present study (page 6, lines 13-19).

Minor:

Page 4, last line of intro - incomplete sentence

Thanks for the careful review. We have completed the sentence “the combination of two antibodies specific for distinct antigenic sites on the G protein and able to broadly neutralize both RABV and non-RABV lyssaviruses...” that is now replaced with “the combination of two antibodies that bind to different antigenic sites on the RABV G protein and are able to broadly neutralize both RABV and non-RABV lyssavirus isolates, will significantly reduce the risk of PEP failure.” (page 4, lines 27-29).

Page 7 - last paragraph. It does not appear to me that CR57 has a greater range of IC50/90 values than RVC20, though undoubtedly it is less potent.

We agree with the Referee's comment and have changed the text accordingly by restricting the comment about the broader range of IC50s to the comparison between RVC58 vs RAB1 and CR4098. This sentence has been rewritten (page 8, lines 7-8) as follows: "CR4098 and RAB1 showed a broader range of IC50/IC90 values (0.7-23600 ng/ml, 1-4153 ng/ml, respectively), neutralizing six and three RABV isolates, respectively, with IC50 >1000 ng/ml, a concentration which is likely not to be effective in PEP."

Figure 5B is extremely hard to follow - the addition of a schematic showing how the chimeras are constructed might be useful to clarify this.

We agree with Referee's comment and acknowledge him for the suggestion to include a diagrammatic sketch showing how the chimeras were constructed. We have therefore included a schematic showing generation of epitope swapped G protein in the new **Figure EV1**.

Corresponding Author Name: Davide Corti
 Journal Submitted to: Embo Molecular Medicine
 Manuscript Number: